# Optimization of γ-Aminobutyric Acid Production in Brown Rice via Prolonged Seed Priming

**DOI:** 10.3390/plants13243594

**Published:** 2024-12-23

**Authors:** Lingxiang Xu, Xiaoan Wang, Qixiang Li, Yuqing Niu, Guohui Ding, Jiawei He, Weiping Chen, Dagang Tian

**Affiliations:** 1Fujian Laboratory for Rice Germplasm Innovation and Molecular Breeding, Biotechnology Research Institute, Institute of Crop Sciences, Fujian Academy of Agricultural Sciences, Fuzhou 350001, China; 2Academy for Advanced Interdisciplinary Studies, Plant Phenomics Research Center, College of Agriculture, Nanjing Agricultural University, Nanjing 210095, China; 3Sanming Agricultural Bureau, Sanming 365000, China

**Keywords:** routine priming, prolonged priming, germinated brown rice, γ-aminobutyric acid, seed integrity, desiccation tolerance

## Abstract

Germinated whole seeds possess elevated levels of bioactive nutrients; however, their application is hindered by several constraints. The germination process is typically time-consuming, and germinated seeds present challenges in terms of storage and transportation compared to dry seeds. This study introduces a novel processing method for rice, termed prolonged priming (PLP), aiming to combine the benefits of germinated and dry seeds. PLP involves soaking the seeds until the embryo exposure stage, followed by redrying. At 10 h (hour) germination post PLP, the γ-aminobutyric acid (GABA) levels in Hanyou73 (HY73) and IRAT exceeded 20 mg/100 g. Additionally, there was an induction of various nutrient components, including an increase in protein content, a reduction in amylose levels, and an elevation in fatty acid content, among others. Malondialdehyde levels, indicating oxidative damage, remained stable, and PLP preserved better seed integrity compared to routine priming in the desiccation-tolerant HY73. Collectively, the PLP treatment demonstrates an optimization of the nutritional value and storage in germinated brown rice (GBR). This novel process holds potential for enhancing the nutritional profile of GBR and may be applicable to other crop species.

## 1. Introduction

Germination is a common method for producing bioactive compounds in seed crops, especially cereals and beans [1,2,3]. These compounds, including γ-aminobutyric acid (GABA), vitamins B and E, and trace elements, rely heavily on seed germination [4]. During rice seed germination, nearly all essential amino acids for humans are enriched, except glutamic acid, which is a precursor of GABA [5,6,7]. White rice raises blood glucose, whereas GABA lowers it. Germinated brown rice (GBR) sees a significant rise in GABA, which has anti-diabetic and anti-Alzheimer’s effects [7,8,9]. GABA regulates glucose, reduces hyperglycemia and hypertension, and acts as an inhibitory neurotransmitter to lower anxiety [10]. Compounds such as Gama-Oryzanol and selenomethionine are vital for seed nutrition [11], with germination playing a crucial role in converting inorganic to organic selenium [12,13].

However, the practical use of germinated seed products is limited by their short shelf life and susceptibility to deterioration, akin to fresh vegetables. While redrying or parboiling can extend their storage time, these methods cause nutrient loss [14]. Re-dried GBR is also impractical due to fragile shoots that break during handling and the lengthy process (over 20 h (hour)) required to reach significant nutrient levels, making it unsuitable for quick preparation [7,12]. Additionally, inconsistent seed quality from poor germination conditions further complicates their use.

Speeding up seed germination (known as “priming”) could address the challenges mentioned. Seeds, which have a long shelf life as seeds, can be quickly germinated if their germination process is significantly accelerated. Rice and oat products can be utilized before the sprouting stage with minimal embryo growth (e.g., in porridge, beer, rice bubbles, rice milk), making the process less time-consuming [15]. This approach also eliminates the need to separate roots from shoots/seeds before radicle protrusion [16].

This study presents prolonged priming (PLP), an innovative method that enhances traditional seed priming by extending the soaking duration to overcome the limitations of conventional germination techniques. Seeds are redried at the embryo exposure (dehiscent) stage, a “semi-germination” phase, where they exhibit significant desiccation tolerance (DT), enabling storage and subsequent germination. Additionally, redried PLP seeds, similar to pre-harvest germinated seeds, are better for variety tests because GABA levels are affected by germination, not just genotypes, making them more suitable for seed quality classification than dry seeds or rice sprouts (Appendix A). GABA, associated with germination, is the primary bioactive product in this processing method. The imbibition–redrying process, along with routine priming, can rapidly elevate GABA levels in brown rice. When combined with appropriate germination practices and varieties, PLP can effectively leverage germinated seeds.

## 2. Results

### 2.1. PLP Demonstrated Effective Embryo Exposure

The germination percentages of nine indica rice varieties and eight japonica rice varieties ranged from 9.0% to 98.35%, with no significant difference observed between indica and japonica rice varieties (Table 1). As NanGeng46, WuYuGeng39, and NanGeng 9108 showed no embryo protrusion, along with very low germination and desiccation survival rates, these varieties were not selected for further experimentation (Table 1). Typically, IRAT109, Zhaxima (ZXM), Minghui63 (MH63), and Hanyou73 (HY73), recognized as brown rice, were utilized to assess their seed germination across various treatments. Results indicated that ZXM and HY73 had significantly higher germination rates over both control (CK) and hydro-primed (HP) seeds at 5 h, with these differences becoming more pronounced at 10 h (Figure 1A–C) (*p* < 0.05). In contrast, MH63 and IRAT109 exhibited less enhancement from embryo exposure, yet still outperformed CK and HP at 7 and 10 h (Figure 1C). To summarize, the embryo exposure of these four varieties after PLP were better than CK and HP, and further testing will be conducted.

### 2.2. PLP Increased GABA Synthesis

PLP significantly increased GABA content in seeds compared to CK and HP (Figure 2). For IRAT109, the GABA content in the PLP seeds was gradually up-regulated from 0 to 10 h of soaking, reaching its peak at 10 h (Figure 2). In contrast, although the GABA contents in the PLP seeds of ZXM and HY73 were suppressed at 5 h of soaking, they still reached the highest levels at 10 h of soaking. Furthermore, the GABA levels even exceeded 20 mg/100 g dry weight for IRAT109 and HY73 after 10 h of soaking (Figure 2).

### 2.3. PLP Preserved Better Grain Wholeness and Bran Color than HP

In comparison to PLP, which maintained grain integrity akin to CK, HP significantly reduced both the husked rice yield (Y1) and the percentage of intact brown rice (Y2) in IRAT109 and MH63 (Figure 3A,B). Integrity in this context refers to husked rice yield, which is the percentage of rice remaining after the husk is removed from paddy rice, as well as the percentage of intact brown rice, meaning the proportion of whole, unbroken kernels of brown rice. Moreover, the hardness of the seed of HY73 in PLP following 10 h of soaking was higher than HP (Figure 3C). We speculate that PLP avoided the disadvantages of HP, as dry GBR shoots are susceptible to breaking during packaging or transport if they grow too long.

Furthermore, a comparative analysis of the pigmentation in ZXM and Xiaohong was conducted between HP and PLP conditions. Samples of ZXM-PLP and ZXM-HP were selected from 88 and 64 seeds, respectively, while Xiaohong-PLP and Xiaohong-HP were selected from 61 and 39 seeds, respectively. The results indicated that the red bran of both ZXM and Xiaohong exhibited increased darkening following PLP treatment. Notably, ZXM-PLP demonstrated a significantly darker pigmentation compared to ZXM-HP in the bran (Appendix A).

### 2.4. Benefits of PLP on Nutrient Levels

After PLP, dry seeds (with husk) had significantly higher protein and lower amylose content, at 9.56% and 15.18%, respectively. In contrast, the corresponding values in CK were 9.07% and 18.44%, respectively. Fatty acid content increased, albeit not significantly, with 15.31% in PLP and 14.83% in CK (Figure 4A).

Based on the research findings, the drought-sensitive MH63 and drought-tolerant HY73 exhibited differing responses to HP treatments, which resulted in significant variations in malondialdehyde (MDA) levels across various soaking durations. In contrast, PLP treatments maintained relatively consistent MDA levels at soaking durations of 3, 10, and 15 h, with the lowest MDA level observed at 10 h (Figure 4B). Therefore, a soaking duration of 10 h appears to be advantageous.

## 3. Discussion

### 3.1. Feasibility of PLP and Short Germination for GBR Production

The key to combining the advantages of GBR (nutrient-rich) and storability lies in achieving rapid germination. Routine hydro-priming enhances seed germination, which can easily break during packaging or transport if the shoot elongates excessively [17,18]. In contrast, PLP seeds are versatile and can be used with or without germination (Figure 5A, Appendix A). As dry seeds, PLP grains can be used in porridge, added to wheaten foods or beers to boost GABA content, and serve as a gluten-free alternative [19]. Thus, PLP not only avoids the drawbacks of routine priming but also the production of routine dry GBR. Moreover, PLP allows for predicting poor seed quality through low embryo protrusion percentages, helping to avoid deterioration during germination.

This study measured GABA content using the dry mass of non-germinated seed, unlike previous research that used germinated seed biomass, which can lose weight through respiration and cause bias. Using dry mass offers a more accurate measure of GABA levels. Furthermore, the study confirmed that germination enhances GABA synthesis only after a certain threshold (5 h was insufficient for PLP seeds in this study). GABA was concentrated in the coleoptile (shoot) of GBR [20]. Therefore, rapid seed germination/shoot elongation is crucial for surpassing this threshold, especially under sub-optimal conditions where even a second-best germination is sufficient [21]. This method may also apply to Zizania rice (*zizania palustris* L.) or eelgrass (*Zostera marina* L.), which is a highly nutritious food, rich in protein, dietary fiber, vitamins, and minerals, with a low glycemic index, and low fat content [22].

### 3.2. PLP Induces Multiple Bioactive Nutrients

The study found that PLP optimizes the nutritional profile of germinated brown rice by significantly increasing GABA and protein content, reducing amylose levels, and enhancing the stability of various nutrients. Previous studies showed that the increase in GABA, protein, fatty acids, and nutrients like vitamins and essential amino acids mainly occurs in the embryo during germination [23]. During brown rice germination, amylose degrades while protein and fatty acid synthesis occurs, as shown by Garg et al. (2021) [4]. This process is fundamental to seed germination and involves various nutrient changes. The importance of multi-nutrients suggests that germinating seeds can act as plant bioreactors.

During germination, GABA is converted from glutamic acid [24]. Compounds like selenomethionine are also crucial for seed nutrition [25]. While not as plant-dependent as GABA, germination is key in converting inorganic selenium to organic selenium [11,12]. A detailed study of nutrient synthesis during seed germination is recommended to enhance seed value. Additionally, GBR bran, particularly colored varieties such as purple or black, is rich in phenols and flavonoids, indicating the presence of anthocyanins [26]. Although less attention is given to nutrients in the bran compared to the embryo, they also increase during germination [27]. Rice bran can produce aromatic compounds [20,28]. Furthermore, PLP helps preserve pigments. Rapid germination enhances the concentration and synthesis of phenolic acids and other antioxidants in the embryo [29]. The link between morphological changes and metabolite biosynthesis during germination could serve as a useful dynamic phenotype.

### 3.3. PLP Requires a Desiccation-Tolerant Variety

The benefits of priming decrease over time, and the longevity of primed seeds is reduced during storage, limiting its application and indicating that screening for more tolerant varieties for priming is of great importance. To ensure seed quality after the “embryo protrusion” or “embryo exposure” stage, this article suggests using DT varieties, strictly controlling the degree of seed germination, and removing germinated seeds (In industrial environments, it’s crucial to use drought-resistant varieties since their seeds retain quality after re-drying. It’s also important to manage the proportion of germinated seeds to ensure it isn’t excessive or to verify germination). A key step in maintaining the quality of dehydrated seeds is controlling the water potential. In our tests, filter paper is used, while in large-scale operations, gauze is employed to ensure that cells are not damaged by osmotic pressure. Additionally, the rice husk is still present at this stage, which helps with uniform water absorption. In this study, HY73 maintained high GABA production after six months of storage at 4 °C (Figure 2), and all four examined varieties (IRAT109, ZXM, MH63, and HY73) exhibited higher GABA levels with PLP; the degree of increase varied. For instance, IRAT109 showed a gradual up-regulation of GABA content from 0 to 10 h of soaking, consistently outperforming both the control and hydro-primed treatments. Similarly, HY73 and ZXM also significantly increased in GABA levels, particularly at the 10-h mark. Conversely, MH63 exhibited limited GABA levels with PLP for dry seeds and, despite high GABA after 10 h of germination, may accumulate hazardous by-products due to low seed quality (12% germination percentage compared to 30.3% in HY73). Nipponbare, which has poorer DT than MH63 (Table 1), also showed low GABA levels even at a 30.9% germination rate. Nangeng46, used as brown rice, had even worse DT and was excluded from consideration for PLP application. Thus, DT varieties in particular may be promising for high GABA accumulation during PLP. High DT in seeds with emerging shoots or roots is essential to preserve their sprouting ability. Without strong DT, early germinating seeds lose vigor. Therefore, DT rice with PLP appears to be a promising strategy to ensure high GABA content without significant loss of seed vigor during storage (Appendix A, Figure 6). This feasibility for months-long storage could help popularize GBR. Furthermore, DT is influenced by multiple factors beyond genotype, necessitating further research on varieties such as 9311, Hanhui 3, Zhenshan 97, Nagina 22, and LongHuaMaoHu, considering different timings and conditions for their PLP processes.

## 4. Material and Methods

### 4.1. Materials and Processing

Nine representative indica rice varieties and eight japonica rice varieties were selected to evaluate their germination desiccation survival at 30 h after PLP treatment. Typically, HY73 (indica subspecies, drought resistant with upland rice genetic background) is known to be drought resistant and is commercialized, with good flavor [30]. MH63 (indica subspecies, paddy) is a common drought-sensitive variety. MH63 and HY73 seeds were harvested in 2020; IRAT109 (japonica subspecies, upland) and ZXM (japonica subspecies with red bran, upland) seeds were harvested in 2021. All seeds were provided by Shanghai Agrobiological Gene Center.

The processing of rice seeds is shown in Figure 5. Seeds were subjected to two priming methods to advance them to the embryo exposure stage. In the experimental priming method, 5 g of seeds were placed between two layers of filter paper within 13 cm × 13 cm lidded boxes, using 13 mL of pure water at a temperature of 28 °C. Conversely, the mass priming method involved approximately 120 g of seeds wrapped in gauze, soaked with 260 mL of tap water at ambient room temperature (about 25 °C). A notable distinction between these methods is that mass priming lacks precise conditioning, allowing for the collection of sprouted seeds (Figure 5B). The seeds were subsequently collected at approximately 6 h intervals during the imbibition process, commencing with the initial collection (Table 2). Following collection, the seeds were subjected to a redrying process at 35 °C in a drying apparatus for a duration of 8 h to restore them to their approximate original water content. Notably, the IRAT109 variety required a longer drying period of ten hours due to its relatively larger seed size compared to other varieties.

### 4.2. Evalution of Embryo Exposure and DT

The criteria for identification of seed embryo exposure are shown in Figure 6. Seeds in this study were collected once their white embryo became visible (in Chinese, embryo exposure can be named “white exposure”). In contrast to routine germination processes, the germination of brown rice is distinguished by the protrusion of the plumule to a length of 0.5–1 mm, with the radicle generally not emerging before the plumule (Figure 6E). DT was assessed by evaluating seed survival following PLP, at which point the coleoptile length should range from 0.5 mm to 1 mm, occurring just prior to the full germination of the brown seed [31] (Figure 6E).

### 4.3. Examination of Brown Rice’s Physical Characteristics

Husked rice yield (Y1) was calculated as follows:Y1 = (m1 + m2)/2/m. 
where m was the original fresh weight (FW) of ~2 g rice seeds; m1 was the FW of intact brown rice seeds; m2 was the FW of broken rice seeds.

Intact seed percentage (Y2) was calculated as follows:Y2 = m1/(m1 + m2).

The hardness of the brown rice seeds was measured using a texture analyzer (FTC) equipped with a 2 mm-diameter probe. Photographs of the red rice bran from ZXM and Xiaohong were taken inside a photography box in a well-lit room using a cellphone camera (Galaxy A8s, Samsung, Suwon-si, Republic of Korea). The bran color of seeds captured in the same photograph allows for direct comparison.

### 4.4. Analysis of Brown Rice’s Bioactive Nutrients

The quantification of GABA content was conducted using high-performance liquid chromatography (HPLC) equipped with a Thermo Syncronis C18 column (Thermo Fisher Scientific, Waltham, MA, USA; 2.1 mm, 50 mm, 1.7 µm), employing an o-phthalaldehyde (OPA) pre-column derivatization method as described in [32]. The OPA reagent was prepared by dissolving 10 mg of OPA in 2 mL of borate buffer (0.4 M, pH 10.2) containing 30 μL of mercaptoethanol. Both the OPA and GABA solutions were filtered through a 0.45 μm membrane prior to mixing (100 μL of OPA solution with 20 μL of GABA solution), allowing the reaction to proceed for approximately 2 min. Subsequently, 10 μL of the derivatized solution was injected into the HPLC system, maintained at 40 °C. The mobile phase consisted of two components: Mobile Phase A, comprising sodium acetate (25 mM, pH 5.90), and Mobile Phase B, acetonitrile. Flow was set to 0.5 mL/min and the following gradient elution was used: 0 min (10% B), 16 min (56% B), 20 min (56% B), 22 min (10% B), 22 min (10% B). Fluorescence emission and excitation were set to 332 and 425 nm, respectively.

The MDA content of HY73 was quantified using the Jiancheng MDA kit (48T, Plant Malondialdehyde Assay Kit, Colorimetric Method, Nanjing Jiancheng Bio-engineering Institute of production, Nanjing, China). This assay is based on the reaction of MDA with thiobarbituric acid, producing a red chromogen with a maximum absorbance at 532 nm. Approximately 0.7 g of brown rice seeds were ground, and 420 mL of normal saline solution was added to dissolve the MDA. Subsequently, 0.1 mL of this solution was introduced into the reaction system. The mixture was then heated in a water bath at 95 °C for 40 min, followed by centrifugation at 9000× *g* for 10 min. The optical density was measured using a microplate reader. The soluble protein content, amylose content, and fatty acid levels of HY73 were quantified using an infrared analyzer (DA7205) equipped with the brown rice module. For each replicate, approximately 20 g of dehulled seeds were placed on the testing plate. Prior to analysis, seeds that were rotten, indicated by a black discoloration rather than the expected white or slightly brown appearance, as well as any broken seeds, were excluded from the sample.

### 4.5. Statistical Analysis

All experiments were repeated independently at least three times. Data collected were statistically analyzed by one-way analysis of variance followed by least significant difference test at *p* = 0.05 using DPS software (v18.10).

## 5. Conclusions

This study evaluated a novel seed processing method, PLP, which optimizes the nutritional profile of germinated brown rice by significantly increasing GABA and storage of GBR, enhancing protein content, and reducing amylose levels. This method not only boosts the health benefits associated with GBR but also tackles practical challenges related to seed quality classification before germination, as embryo protrusion indicates seed viability, and storage. Combining PLP with short-term germination and desiccation-tolerant varieties facilitates overnight germination while preserving the nutrient value and storability of rice seeds.

## Figures and Tables

**Figure 1 plants-13-03594-f001:**
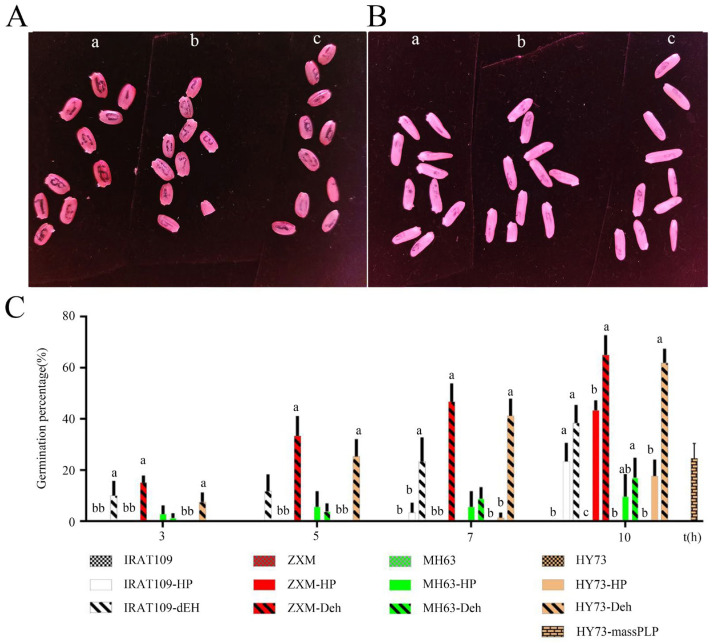
Comparative analysis of seed germination among different treatments. (**A**) Comparative analysis of ZXM under different treatments: (**a**) PLP; (**b**) hydro-primed seeds; (**c**) control. (**B**) Comparative analysis of HY73 under different treatments: (**a**) PLP; (**b**) hydro-primed seeds; (**c**) control. (**C**) Comparative analysis of germination percentage among different varieties under different treatments. Different letters in (**C**) indicate significant difference (*p* < 0.05). Bars indicate standard errors. Each sample contained 8 to 10 replicates (10 seeds per replicate).

**Figure 2 plants-13-03594-f002:**
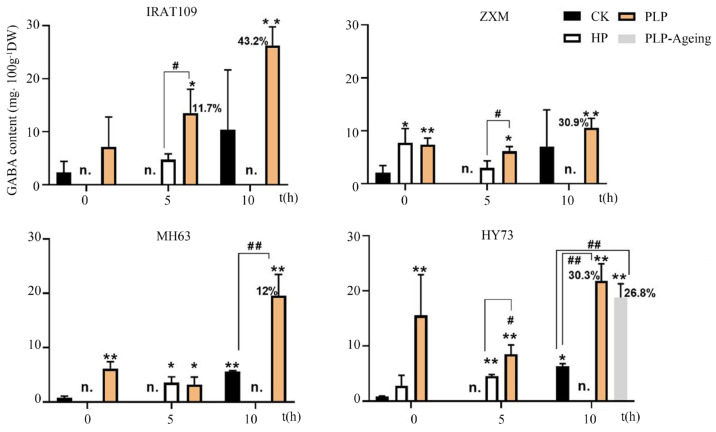
The comparison of GABA contents in IRAT109, ZXM, MH63, and HY73 during different germination processes show that the highest levels of GABA were found in PLP at 10 h. The soaking conditions were 10 h soaking, and soaking temperature at 35 °C. GABA contents were determined at 0, 5 and 10 h of soaking. Data are mean ± SD (n = 3). The *, ** indicate significant (*p* < 0.05, *p* < 0.01, respectively) increase of GABA compared with the control. #, ## indicate significant (*p* < 0.05, *p* < 0.01, respectively) increase of GABA at 5 h or 10 h of soaking compared with other treatments. n., non-pretreated, non-soaked seeds. CK: Control treatment. HP: hydro-primed, a technique in which the plant seeds are presoaked in water at an optimal temperature for a specified duration, followed by a natural drying process that returns them to their initial seed weight. PLP-ageing: HY73 PLP seeds stored at 4 °C for 4 months.

**Figure 3 plants-13-03594-f003:**
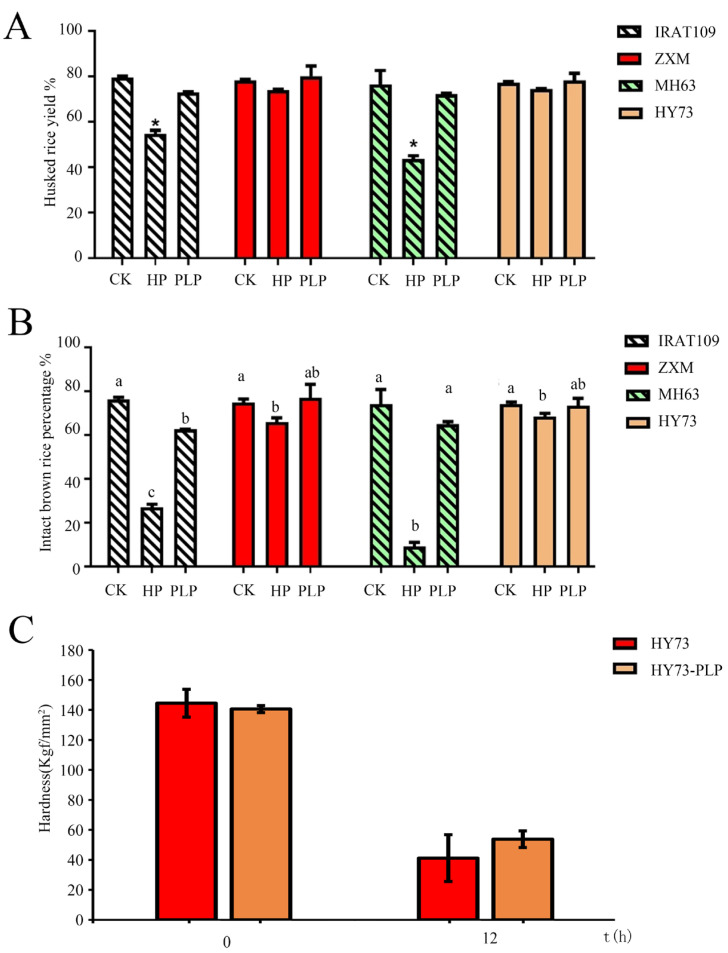
Comparative analysis of rice husk yield, percentage of whole brown rice and hardness among different treatments. (**A**) Husked rice yield. HP notably decreased husked rice yield. (**B**) Seed intactness comparison. Hydropriming resulted in significantly lower husked rice yield than other treatments. (**C**) Seed hardness comparison. Bars indicate standard errors. Each sample contained two or three replicates. Data are mean ± SD (n = 3). The * indicates significance (*p* < 0.05). Different letters indicate significant difference (*p* < 0.05).

**Figure 4 plants-13-03594-f004:**
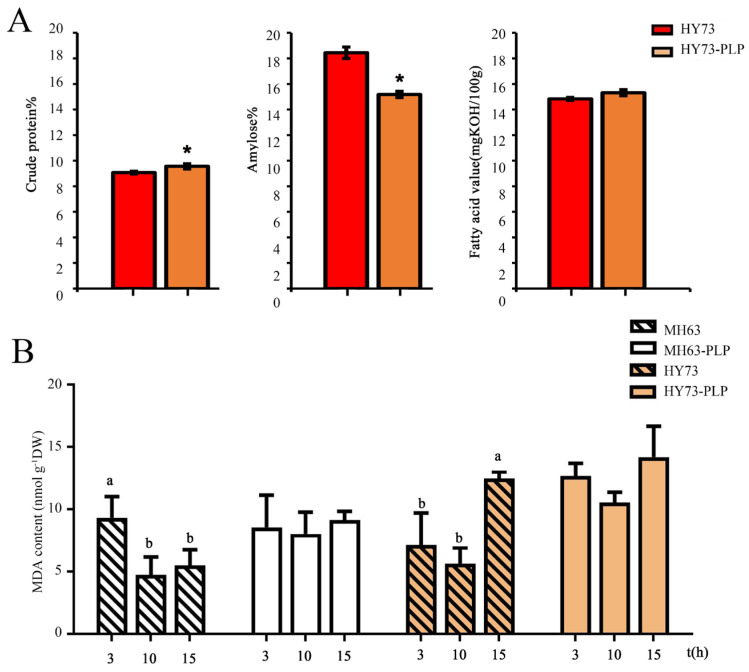
Impact of soaking time on nutrient levels. (**A**) The level of HY73 control seeds (three replicates) was significantly different from PLP (four replicates) seeds. (**B**) MDA level with different soaking durations within the same variety and the same pre-treatment. Each sample contained three replicates. Different letters and * indicate significant difference (*p* < 0.05).

**Figure 5 plants-13-03594-f005:**
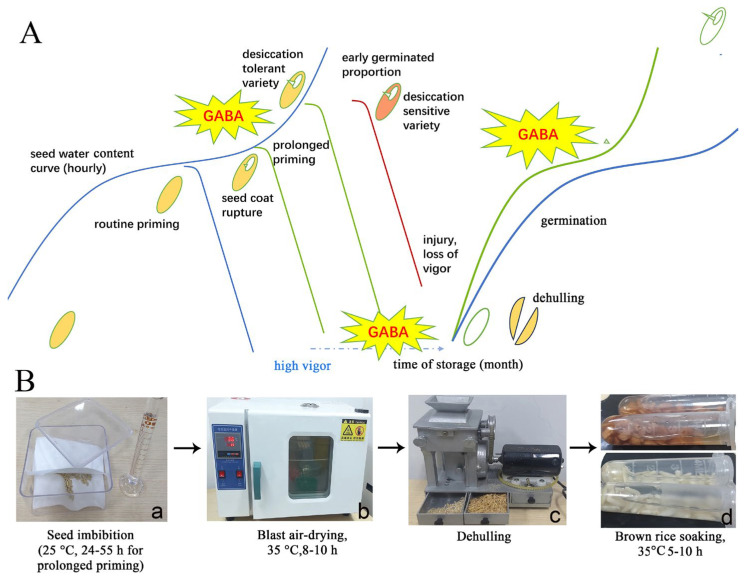
Graphic summary of tillage with extended priming and drought-resistant seed and experiments on seed peeling. (**A**) Graphical summary of extended priming and brief germination using drought-resistant seeds. (**B**) Experimental dehulling of seeds. (**a**) Measuring the water volume as a preparation for germination, seeds were then subjected to imbibition and collected once embryo exposure was detected, from 24 to 55 h of germination; (**b**) blast air-drier for seeds in gauze; (**c**) a JLGL45 husk machine (JLGJ45-B, Guangzhou Hu Ruiming Instrument Co., Ltd. Guangdong, China), which peels the seeds and separates the brown rice seeds; both intact and broken seeds were collected and then weighed; (**d**) brown rice seeds were then incubated in 2 mL centrifugal tubes.

**Figure 6 plants-13-03594-f006:**
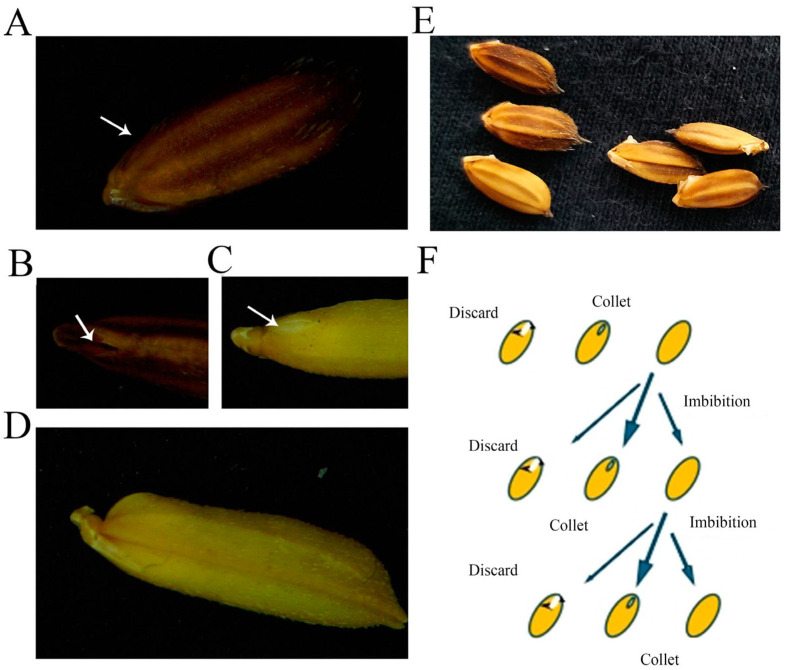
Standards for detecting seed embryo exposure. (**A**) Front view of embryo exposure in ZXM. (**B**) Flank view of embryo exposure in ZXM. (**C**) Front view of dehiscence in HY73. (**D**) Flank view of dehiscence in HY73. The arrows indicate the tiny dehiscence with exposed white embryo. (**E**) The left three seeds represent the accepted very tiny protrusion (also for DT test), and the right three ones were unaccepted for seed preparation for physio-chemical studies. The accepted criteria for the DT test is the protrusion of the plumule to a length of 0.5–1 mm. (**F**) The process of collecting seeds with embryo exposed.

**Table 1 plants-13-03594-t001:** Survival-based desiccation tolerance in seeds with shoot protrusion.

Indica	Germination Percentage	Standard Error	Japonica	Germination Percentage	Standard Error
HY73	72.97%	2.77%	IRAT109	81.97%	4.48%
MH63	61.88%	9.82%	ZXM	50.45%	8.46%
Nagina22	98.35%	0.84%	YunLu8	81.92%	9.32%
9311	94.96%	0.99%	LongHuaMaoHu	95.11%	0.83%
ZhenShan97B	95.45%	1.66%	NanGeng46	11.74%	5.48%
HanHui3	93.62%	8.33%	Nipponbare	39.14%	11.53%
HanHui15	85.76%	3.91%	WuYuGeng39	9.07%	3.58%
HuangHuaZhan	76.14%	4.95%	NanGeng9108	20.22%	2.99%
HuHan1B	22.93%	6.00%			

**Table 2 plants-13-03594-t002:** Timing for seed collection.

Round of Collection	1st	2nd	3rd	4th
IRAT109	24 h	30 h	36 h	
ZXM	30 h	36 h	42 h	48 h
MH63	27 h	36 h	42 h	
HY73	28 h	36 h	42 h	

Seeds were soaked on two pieces of wet filter paper at 28 °C. A few seeds with protruded roots or shoots longer than 2 mm were abandoned. Before the last round of collection, the embryo-exposure percentage was >80%.

## Data Availability

The data presented in this study are available upon request from the corresponding author.

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
