# Peer review of "Optimization of γ-Aminobutyric Acid Production in Brown Rice via Prolonged Seed Priming"

_plants, 2024, doi:10.3390/plants13243594_

Round 1
Reviewer 1 Report (Previous Reviewer 2)
Comments and Suggestions for Authors
Reviewer report of the 2nd submission
The manuscript: Prolonged Priming Enhances γ-Aminobutyric Acid Production in Germinated Brown Rice was resubmitted after publication denied by the editor.
I was advising the publication after my last review since the authors successed to greatly improve the initial manuscript but still remaining with possibilities to be improved in several sections. I consider manuscript ok to be accepted after the authors success to do the corrections suggested by the end of this review, but would like they took some time to improve a little bit more the discussion and conclusions.
Authors did not correct all the suggested corrections of the last revision (the ones concerning the equations).
Title was improved with the correction made.
Abstract, introduction, discussion, M&M, conclusions are smaller and easy to understand, better connected.
Line 35 A space is missing between GABA and the [5, 6, 7]
Line 67: is it timing or priming? I did not understand what the authors want to say
Line 163: the figure 5 need to be simplified and better explained. I suggest to use time of storage (month) in spite of t(month) storage; the pictures need to have the letters a), b), c) and d) before their legends in the figure 5.
In lines 262 and 266 equation number are still missing
and I still have doubt about equation Y1, is it like it is: Y1=(m1+m2/2)/m or is it Y1= (m1+m2)/2/m?
line 292: g should be in italic
line 301: is analysis and not analy-sis
line 302: p should be a small letter and in italic
line 359: reference number 11. What is it proteins[J]?
n Rice
Author Response
Comments 1: Line 35 A space is missing between GABA and the [5, 6, 7]
Response 1:Thanks for the reviewer’s suggestion.
This has been edited.
Comments 2: Line 67: is it timing or priming? I did not understand what the authors want to say
Response 2:It’s has been deleted.
Comments 3: Line 163: the figure 5 need to be simplified and better explained. I suggest to use time of storage (month) in spite of t(month) storage; the pictures need to have the letters a), b), c) and d) before their legends in the figure 5.
Response 3:The figure 5 has been edited
Comments 4: In lines 262 and 266 equation number are still missing
and I still have doubt about equation Y1, is it like it is: Y1=(m1+m2/2)/m or is it Y1=
(m1+m2)/2/m?
Response 4:They have been added in line 97
The equation has been edited.
Comments 5: line 292: g should be in italic
Response 5:It has been edited.
Comments 6: line 301: is analysis and not analy-sis
Response 6:It has been edited.
Comments 7: line 302: p should be a small letter and in italic
Response 7:It has been edited.
Comments 8: line 359: reference number 11. What is it proteins[J]?
Response 8:It has been edited.
Reviewer 2 Report (New Reviewer)
Comments and Suggestions for Authors
The manuscript "Prolonged Priming Enhances γ-Aminobutyric Acid Production in Germinated Brown Rice" contributed by Lingxiang Xu, Xiaoan Wang, Qixiang Li, Yuqing Niu, Guohui Ding, Jiawei He, Weiping Chen, and Dagang Tian, introduces prolonged priming (PLP) as a novel method to enhance GABA production in germinated brown rice. PLP improves GABA levels and nutrient content by optimizing soaking and redrying conditions while maintaining seed integrity. The findings provide a promising approach to increasing germinated brown rice's nutritional value and storability. After analysis I verify that:
1) The current title is appropriate as it communicates the study's central focus on "prolonged priming" and its effect on GABA production. However, it can be refined for conciseness and better clarity. Then, I suggest Optimization of GABA Production in Brown Rice via Prolonged Seed Priming.
2) The English must be improved because there are many tipos and grammatical errors. Give examples: In the abstract, replace "which aims to integrate the advantages of both germinated and dry seeds" with "aiming to combine the benefits of germinated and dry seeds" for conciseness. "Unlike conventional priming, PLP entails soaking the seeds" with "PLP involves soaking the seeds" (simpler phrasing). Results Section: "ZXM and HY73 exhibited significant higher germination rates" with"significantly higher germination rates." Ensure uniformity in terminology: Use either "GABA" or "γ-aminobutyric acid" consistently throughout the manuscript.
3) The introduction is well-written and provides an adequate background on GABA synthesis and germination. However the novelty of the study should be explicitly highlighted earlier. For example: "This study introduces PLP as an innovative solution to address the limitations of traditional germination methods." The linkage between prolonged priming and GABA enhancement requires clearer justification supported by references.
4) Figures and Tables Quality. Figure 1: Lacks detailed labeling for clarity. Add sub-panel titles (e.g., A, B, C) and ensure font uniformity. Figure 2: Improve legend readability; explain acronyms such as "HP" and "PLP" clearly in the figure caption. Table 1: "Standard Error" column formatting needs improvement (center alignment). Missing data (e.g., significance annotations) in a few cells should be clarified.
5) Errors and omissions on Materials and Methods. Reagents: Missing complete information on the manufacturer (company, city) and part numbers for reagents like "Jiancheng MDA kit" and "Thermo Syncronis C18 column." Example: "Thermo Fisher Scientific, Waltham, MA, USA; Part #XXXX."
Equipment: The "JLGL45 husk machine" lacks proper manufacturer information. Provide the model, manufacturer name, and location.
6) Results are presented clearly but lack sufficient descriptive statistics for replication. Example: For "PLP significantly increased GABA content," specify p-values and effect sizes. Figures effectively complement the data but should include explicit figure captions for non-expert readers.
7) The discussion is generally satisfactory but: Does not address limitations comprehensively. Suggested inclusions: Effects of prolonged priming on different rice cultivars beyond the tested varieties. Limitations regarding seed storage conditions and long-term GABA stability. Comparative analysis with existing studies is minimal. Consider integrating studies on GABA synthesis and PLP benefits from other crops for broader context.
8) Bibliographic Reference Analysis. Only 20% of the references was published after 2020. The ideal must be 40%. Please update it
Based on the evaluation: Strengths: Novel concept of prolonged priming; clear experimental results supporting GABA production. Weaknesses: Minor omissions in methodology, lack of detailed statistical reporting, and incomplete citations for reagents/equipment.
Recommendation:
- The manuscript can be accepted for publication with Minor Revisions
Must be improved by a native English speaker
Author Response
Comments 1: The current title is appropriate as it communicates the study's central focus on "prolonged priming" and its effect on GABA production. However, it can be refined for conciseness and better clarity. Then, I suggest Optimization of GABA Production in Brown Rice via Prolonged Seed Priming.
Response 1: Thanks for the reviewer’s suggestion.
The title has been edited according to your suggestion.
Comments 2: The English must be improved because there are many tipos and grammatical errors. Give examples: In the abstract, replace "which aims to integrate the advantages of both germinated and dry seeds" with "aiming to combine the benefits of germinated and dry seeds" for conciseness. "Unlike conventional priming, PLP entails soaking the seeds" with "PLP involves soaking the seeds" (simpler phrasing). Results Section: "ZXM and HY73 exhibited significant higher germination rates" with"significantly higher germination rates." Ensure uniformity in terminology: Use either "GABA" or "γ-aminobutyric acid" consistently throughout the manuscript.
Response 2:The manuscript has been edited based on your suggestion except that use either "GABA" or "γ-aminobutyric acid" consistently throughout the manuscript as we use GABA as an abbreviation for γ-aminobutyric acid.
Comments 3: The introduction is well-written and provides an adequate background on GABA synthesis and germination. However the novelty of the study should be explicitly highlighted earlier. For example: "This study introduces PLP as an innovative solution to address the limitations of traditional germination methods." The linkage between prolonged priming and GABA enhancement requires clearer justification supported by references.
Response 3:The manuscript has been edited based on your suggestions.
As this study firstly focused on the connection between prolonged priming and GABA enhancement, no previous study can be referenced.
Comments 4: Figures and Tables Quality. Figure 1: Lacks detailed labeling for clarity. Add sub-panel titles (e.g., A, B, C) and ensure font uniformity. Figure 2: Improve legend readability; explain acronyms such as "HP" and "PLP" clearly in the figure caption. Table 1: "Standard Error" column formatting needs improvement (center alignment). Missing data (e.g., significance annotations) in a few cells should be clarified.
Response 4:The manuscript has been edited based on your suggestions.
The lack of significant annotations is mainly due to the absence of comparative data.
Comments 5: Errors and omissions on Materials and Methods. Reagents: Missing complete information on the manufacturer (company, city) and part numbers for reagents like "Jiancheng MDA kit" and "Thermo Syncronis C18 column." Example: "Thermo Fisher Scientific, Waltham, MA, USA; Part #XXXX."
Equipment: The "JLGL45 husk machine" lacks proper manufacturer information. Provide the model, manufacturer name, and location.
Response 5:These information has been added based on your suggestions.
Comments 6: Results are presented clearly but lack sufficient descriptive statistics for replication. Example: For "PLP significantly increased GABA content," specify p-values and effect sizes. Figures effectively complement the data but should include explicit figure captions for non-expert readers.
Response 6:The manuscript has been revised
Comments 7: The discussion is generally satisfactory but: Does not address limitations comprehensively. Suggested inclusions: Effects of prolonged priming on different rice cultivars beyond the tested varieties. Limitations regarding seed storage conditions and long-term GABA stability. Comparative analysis with existing studies is minimal. Consider integrating studies on GABA synthesis and PLP benefits from other crops for broader context.
Response 7:The discussion has been edited based on your suggestions.
Comments 8: Bibliographic Reference Analysis. Only 20% of the references was published after 2020. The ideal must be 40%. Please update it
Response 8:The reference has been adjusted to meet your requirement.
Reviewer 3 Report (New Reviewer)
Comments and Suggestions for Authors
This article presents an innovative method for enhancing the nutritional value of brown rice. The manuscript has significantly improved from the first submitted draft, as shown in the track changes.
My minor editorial comment and suggestions as follows:
Line 87: Inconsistent capitalization in variety names (e.g., "HanYou73" vs "Hanyou73")
Line 587-592 Suggestion for Conclusions section: Please add the implications of the results of this study to the conclusion for example "PLP could potentially optimize the nutritional value and storage of GBR." "This method may be applicable to other crop species!" & "PLP allows for seed quality classification before germination, as embryo extrusion indicates seed viability..."

Author Response
Comments 1: Line 87: Inconsistent capitalization in variety names (e.g., "HanYou73" vs "Hanyou73")
Response 1:Thanks for the reviewer’s suggestion
It has been edited.
Comments 2:Line 587-592 Suggestion for Conclusions section: Please add the implications of the results of this study to the conclusion for example "PLP could potentially optimize the nutritional value and storage of GBR." "This method may be applicable to other crop species!" & "PLP allows for seed quality classification before germination, as embryo extrusion indicates seed viability..."
Response 2:The conclusion has been revised based on your suggestions.
Reviewer 4 Report (New Reviewer)
Comments and Suggestions for Authors
The study carried out is of interest to improve the nutritional quality of germinated brown rice, especially by increasing GABA, which is attributed with important beneficial effects on health. The increases that occurred in some cases (20 mg/100 g) are very high when compared with non-germinated rice or other sources of GABA. The proposed prolonged priming method showed results that improved some nutritional aspects of germinated brown rice obtained by traditional priming.
Some of the observations to improve the manuscript are the following:
1. In the introduction, I consider that the paragraph (Page 1 rows 39-43) does not contribute to the topic addressed in this study. What they describe is a topic of interest for another study.
“Compounds like Gama-Oryzanol and selenomethionine are also crucial for seed nutrition [11], germination is key in converting inorganic selenium to organic selenium [12, 13]. A detailed study of nutrient synthesis during seed germination is recommended to enhance seed value. The link between morphological changes and metabolite biosynthesis during germination could serve as a useful dynamic phenotype.”
2. In the introduction, I consider that the paragraph (page 2, rows 64-67) is not clear to me, with regard to the genotype and the production of GABA, I consider that it is information of interest for another study.
“Furthermore, redried PLP seeds resemble pre-harvest germinated seeds and are advantageous for variety tests since GABA levels are influenced by the germination process rather than solely by genotypes, making them more appropriate for seed quality classification than dry seeds or rice sprouts, which complicate timing.”
3 In the introduction, you can present information on studies directly related to the topics addressed, for example, what information has been published regarding the improvement of the nutritional quality of brown rice by routine priming, or germination processes that affect the improvement of the constituents, such as GABA, protein, fatty acids.
4. In Figure 1 in section C, it is not clear what the acronyms mean in each variety, for example, in the variety HY73, which means HY73HP, HY73Deh and HY73-massPLP.
5. In Figure 5, in the text “Seed water cotent curve (hourly)” change to “Seed water content curve (hourly)”
Author Response
Comments 1: In the introduction, I consider that the paragraph (Page 1 rows 39-43) does not contribute to the topic addressed in this study. What they describe is a topic of interest for another study.
“Compounds like Gama-Oryzanol and selenomethionine are also crucial for seed nutrition [11], germination is key in converting inorganic selenium to organic selenium [12, 13]. A detailed study of nutrient synthesis during seed germination is recommended to enhance seed value. The link between morphological changes and metabolite biosynthesis during germination could serve as a useful dynamic phenotype.”
Response 1:Thanks for the reviewer’s suggestion
It has been revised.
Comments 2: In the introduction, I consider that the paragraph (page 2, rows 64-67) is not clear to me, with regard to the genotype and the production of GABA, I consider that it is information of interest for another study.
“Furthermore, redried PLP seeds resemble pre-harvest germinated seeds and are advantageous for variety tests since GABA levels are influenced by the germination process rather than solely by genotypes, making them more appropriate for seed quality classification than dry seeds or rice sprouts, which complicate timing.”
Response 2:It has been revised.
Comments 3: In the introduction, you can present information on studies directly related to the topics addressed, for example, what information has been published regarding the improvement of the nutritional quality of brown rice by routine priming, or germination processes that affect the improvement of the constituents, such as GABA, protein, fatty acids.
Response 3:It has been revised.
Comments 4: In Figure 1 in section C, it is not clear what the acronyms mean in each variety, for example, in the variety HY73, which means HY73HP, HY73Deh and HY73-massPLP.
Response 4:HY73 is CK, no treatment.
Comments 5: In Figure 5, in the text “Seed water cotent curve (hourly)” change to “Seed water content curve (hourly)”
Response 5: It has been revised.
This manuscript is a resubmission of an earlier submission. The following is a list of the peer review reports and author responses from that submission.
Round 1
Reviewer 1 Report
Comments and Suggestions for Authors
Very poorly written
The abbreviated terms are often interchangeably used with the original names, confusing the reader.
The methodology is so confusing and the weakest section. Two types of soaking were described (experimental vs mass priming) without an explanation of what was used for what. Figure 7A shows 24-55 h of soaking, but Table 2 shows 24-48 h.
Figure captions are presented in a way that they are part of the main text, not a caption.
Figure 7 caption is “experimental dehulling of seeds and grains,” which the text has not explained properly; Figures 7A, B, and D show soaking, not dehulling. The authors presented “seeds” and “grains” as two different materials in the figure caption without explaining the difference, and throughout the text, seeds and grains are used interchangeably.
The discussion section is more like the introduction or a review paper. They did not interpret their own results.
Supplementary tables are poorly written and formatted.
It is really hard to follow the workflow of this study.
Please see more comments in the attached pdf.

Author Response
Comment1:The abbreviated terms are often interchangeably used with the original names, confusing the reader.
Response: Thanks for the reviewer’s suggestion.It has been edited in the manuscript.
Comment2:The methodology is so confusing and the weakest section. Two types of soaking were described (experimental vs mass priming) without an explanation of what was used for what. Figure 7A shows 24-55 h of soaking, but Table 2 shows 24-48 h.
Response: The data shown in Table 2 is based on samples taken within the soaking time range defined in Figure 7A. This sampling aims to analyze the impact of different time intervals on the experimental results, allowing for a more accurate assessment of the soaking process's effectiveness. Through this method, we can obtain more detailed experimental data, providing a reliable basis for subsequent research.
Comment3:Figure captions are presented in a way that they are part of the main text, not a caption.
Response: They have been presented in the manuscript.
Comment 4: Figure 7 caption is “experimental dehulling of seeds and grains,” which the text has not explained properly; Figures 7A, B, and D show soaking, not dehulling. The authors presented “seeds” and “grains” as two different materials in the figure caption without explaining the difference, and throughout the text, seeds and grains are used interchangeably.
Response: We have redesigned Figure 7. The study used rice grains for the experiment and it has been edited in the text.
Comment5:The discussion section is more like the introduction or a review paper. They did not interpret their own results.
Response: They have been edited in the manuscript.
Comment6:Supplementary tables are poorly written and formatted.
Response: They have been edited.
Reviewer 2 Report
Comments and Suggestions for Authors
The manuscript is entitled: Prolonged Priming Enhances γ-Aminobutyric Acid Production and Grain Integrity in Germinated Brown Rice.
The manuscript main question is how can the nutritional benefits of germinated grains be optimized while overcoming the practical challenges of germination, storage, and transportation. The paper investigates whether a novel method, called prolonged priming (PLP), can enhance the nutritional profile of rice and improve its practicality in terms of storage and integrity compared to the traditional germination processes. Unlike conventional priming, PLP entails soaking the seeds until the embryo exposure stage, followed by re-drying.
The process of soaking followed by re-drying and routine priming can rapidly increase GABA levels in brown rice. The GABA is linked to germination and is the key bioactive product in this processing method. GABA is well known as a bioactive compound with anti-diabetic properties, among other health actions.
The topic of the text is relevant and addresses a specific gap in the field of grain processing, nutrition, and storage. The research introduces a novel PLP method, which is an extension of the traditional priming processes used in rice. The novelty lies in the fact that it combines the benefits of germination with enhanced storage properties, addressing key challenges in germinated grain applications, such as the time-consuming nature of the process and difficulties in storage and transportation. PLP addresses both the need for increased nutritional content in rice and the practical challenges of processing, storage, and transportation in germinated grains, presenting a meaningful advancement in the field.
Overall, the manuscript is structured, easy to read, but I would like to suggest some corrections, changes and improvements.
The abstract presents overall main studies and used processes with the principal results and conclusion.
The introduction is well-prepared but could be enriched with more information and some minor corrections. Besides GABA there are more bioactive compounds (like Gama oryzanol) that can have their content improved by germination, but the authors only report the GABA.
The results and discussion are well explained, and the results obtained are compared with those of other works, just needing some corrections, especially in the used units. One sentence about indica versus japonica PLP results is missing.
The conclusions need to be improved showing they are consistent with the evidence and arguments presented and answer the main question posed.
The methodology is well described but can be improved with formatting corrections that can be solved and a better description of the used equipment.
The references list needs to be reviewed in terms of format, references are adequate but please insert DOI in bibliographic references to make it easy to the reader to find them. Use the same format for journal names, sometimes is in italic and other times is not. Review from number 11 to the end because they have ref numbers twice. Journals like Foods or Antioxidants don’t need to be followed by (Basel, Switzerland)
The authors presented a section of Figure legends that was important in this phase since their legends were not well formatted. This section must be removed, lines 353 to 406, and this information must be correctly formatted in the figure legends.
Please format correctly all the figure legends with text size 9. To the reader realise the difference between the legend in text size 9 and the text in text size 10.
Don’t use paragraph break in the legends of the figures.
Figures should be listed as: (a) Description of what is contained in the first panel; (b) Description of what is contained in the second panel.
Figures should be placed in the main text near to the first time they are cited.
Please explain Figure 6 in the text, since it is very intricate with lots of lines and text. The t(mo.) storage, is the time of storage in months? If so, use the complete words and not the abbreviations that the reader doesn’t know.
The section list of abbreviations can be used but need to have all the abbreviations you used. OD, WW, HP, CK, Mass PLP, IRAT, NG, Deh,….,Are some of the missing abbreviations.
Considering the comments, I recommend a major revision of the article.
Following are more comments and some specific proposed modifications.
Line 4: after Chen use the 2 in superscript.
Improve paragraph line 30 to 38 writing about other rice bioactive compounds that are expected to increase in content with the rice germination.
Line 31, 32 and 54: need to have a space between the text and the [ref].
Line 54: DT meaning?
Improve the paragraph of line 52 to 59 in order to the reader better understand the research you did, how, why and what improvements to the field do you expect.
Line 62 to 71 first time variety is mentioned should appear with the full name followed by the abbreviation.
Line 73 to 82 is the Figure 1 legend and should be in 9 text size without paragraph break. Please do this for all the figure legends.
Line 78: A and B(grains outside. Please use a space between B and (
Line 80: use the p in italic like in for (p < 0.5) and change it in all the manuscript text
Line 83: Table 1. What is 9311? please use the variety name
Line 88: Figure 2 C and D.
Please use Figure and not Fig. when you refer the figures in the text, review this in all manuscript.
Line 89: remove the big space between PLP and also
Line 94 to 100: is the Figure 2 legend and should be in 9 text size without paragraph break. Please do this for all the figure legends. Please use p in italic for p < 0.05, p < 0.01
Line 100: What is ~4mo..? Please don’t use the sign ~, write the word.
Line 105: (Fig. 2 B,D) change to (Figure 2. B and D) and use the same formatting in all the text
Line 110: use (Figue 3. A).
Line 111: use (Figure 3. B and C).
Lines 115 to 122 are part of Figure 3 legend, so don’t use break between line 114 and 115. “…pigment. (A) Comparison….”
What is ~28º. Do you mean approximately 28 °C?
Material and Methods section: please when refer to equipment use (model, brand, city, country).
Line 321: please add a reference to the HPLC OPA method.
Line 323: an L is missing in the mercaptoethanol quantity, you only have the micro.
Line 327: add the fluorescence emission and excitation wave lengths
Line 333: r/min is it rotation per minute? Usually we use rpm (rotation per minute) or, the more correct way, we use the centrifuge speed in number x g.
Line 330: correct thiobarbituric
Lines 340 and 342: use the equations has the journal format demand in one line with equation numbers
Supplementary section:
Table S1: Please separate PLP from routine priming and no priming, using table lines with border lines.
Table S2: Please separate rice from wheat using another column at the begin and put all the protein results in the same column, for wheat amylose and fatty acid columns leave them without results or use ND (not determined).
Table S3: please correct the hardness unit.
Author Response
Comment1:The introduction is well-prepared but could be enriched with more information and some minor corrections. Besides GABA there are more bioactive compounds (like Gama oryzanol) that can have their content improved by germination, but the authors only report the GABA.
Response: Thank you for your encourage, and some bioactive compounds were mentioned in the introduction.
Comment2:The results and discussion are well explained, and the results obtained are compared with those of other works, just needing some corrections, especially in the used units. One sentence about indica versus japonica PLP results is missing.
Response: We have added to this part of result.
Comment3:The conclusions need to be improved showing they are consistent with the evidence and arguments presented and answer the main question posed.
Response: The conclusions have been rewritten.
Comment4:The methodology is well described but can be improved with formatting corrections that can be solved and a better description of the used equipment.
Response: They have been edited.
Comment5:The references list needs to be reviewed in terms of format, references are adequate but please insert DOI in bibliographic references to make it easy to the reader to find them. Use the same format for journal names, sometimes is in italic and other times is not. Review from number 11 to the end because they have ref numbers twice. Journals like Foods or Antioxidants don’t need to be followed by (Basel, Switzerland)
Response: They have been edited.
Comment6:The authors presented a section of Figure legends that was important in this phase since their legends were not well formatted. This section must be removed, lines 353 to 406, and this information must be correctly formatted in the figure legends.
Response: They have been edited.
Comment7:Please format correctly all the figure legends with text size 9. To the reader realise the difference between the legend in text size 9 and the text in text size 10.
Response: They have been edited.
Comment8:Don’t use paragraph break in the legends of the figures.
Response: They have been edited.
Comment9:Figures should be listed as: (a) Description of what is contained in the first panel; (b) Description of what is contained in the second panel.
Response: We tried it as you suggested
Comment10:Figures should be placed in the main text near to the first time they are cited.
Response: They have been placed in the right place.
Comment11:Please explain Figure 6 in the text, since it is very intricate with lots of lines and text. The t(mo.) storage, is the time of storage in months? If so, use the complete words and not the abbreviations that the reader doesn’t know.
Response: We integrated the graphs to help the readers understand
Comment12:The section list of abbreviations can be used but need to have all the abbreviations you used. OD, WW, HP, CK, Mass PLP, IRAT, NG, Deh,….,Are some of the missing abbreviations.
Response: They have been edited.
Comment13:Line 4: after Chen use the 2 in superscript.
Response: It has been edited.
Comment14:Improve paragraph line 30 to 38 writing about other rice bioactive compounds that are expected to increase in content with the rice germination.
Response: They have been added.
Comment15:Line 31, 32 and 54: need to have a space between the text and the [ref].
Response: They have been edited.
Comment16:Line 54: DT meaning?
Response: It has been added.
Comment17:Improve the paragraph of line 52 to 59 in order to the reader better understand the research you did, how, why and what improvements to the field do you expect.
Response: They have been improved.
Comment18:Line 62 to 71 first time variety is mentioned should appear with the full name followed by the abbreviation.
Response: They have been added.
Comment19:Line 73 to 82 is the Figure 1 legend and should be in 9 text size without paragraph break. Please do this for all the figure legends.
Response: It has been added.
Comment20:Line 78: A and B(grains outside. Please use a space between B and (
Response: It has been added.
Comment21:Line 80: use the p in italic like in for (p < 0.5) and change it in all the manuscript text
Response: They have been edited.
Comment22:Line 83: Table 1. What is 9311? please use the variety name
Response: The variety name is 9311
Comment23:Line 88: Figure 2 C and D.
Response: It has been edited.
Comment24:Please use Figure and not Fig. when you refer the figures in the text, review this in all manuscript.
Response: They have been edited.
Comment25:Line 89: remove the big space between PLP and also
Response: It has been removed.
Comment26:Line 94 to 100: is the Figure 2 legend and should be in 9 text size without paragraph break. Please do this for all the figure legends. Please use p in italic for p < 0.05, p < 0.01
Response: They have been edited.
Comment27:Line 100: What is ~4mo..? Please don’t use the sign ~, write the word.
Response: It has been edited.
Comment28:Line 105: (Fig. 2 B,D) change to (Figure 2. B and D) and use the same formatting in all the text
Response: They have been edited.
Comment29:Line 110: use (Figue 3. A).
Response: It has been edited.
Comment30:Line 111: use (Figure 3. B and C).
Response: It has been edited.
Comment31:Lines 115 to 122 are part of Figure 3 legend, so don’t use break between line 114 and 115. “…pigment. (A) Comparison….”
Response: It has been edited.
Comment32:What is ~28º. Do you mean approximately 28 °C?
Response: It has been edited.
Comment33:Material and Methods section: please when refer to equipment use (model, brand, city, country).
Response: They have been edited.
Comment34:Line 321: please add a reference to the HPLC OPA method.
Response: It has been added.
Comment35:Line 323: an L is missing in the mercaptoethanol quantity, you only have the micro.
Response: It has been edited.
Comment36:Line 327: add the fluorescence emission and excitation wave lengths
Response: It has been added.
Comment37:Line 333: r/min is it rotation per minute? Usually we use rpm (rotation per minute) or, the more correct way, we use the centrifuge speed in number x g.
Response: It has been edited.
Comment38:Line 330: correct thiobarbituric
Response: It has been corrected.
Comment39:Lines 340 and 342: use the equations has the journal format demand in one line with equation numbers
Response: It has been edited.
Comment40:Table S1: Please separate PLP from routine priming and no priming, using table lines with border lines.
Response: It has been edited.
Comment41:Table S2: Please separate rice from wheat using another column at the begin and put all the protein results in the same column, for wheat amylose and fatty acid columns leave them without results or use ND (not determined).
Response: It has been removed.
Comment41:Table S3: please correct the hardness unit.
Response: It has been removed.